# Automatically Generating Visual Hallucination Test Cases for Multimodal Large Language Models

## Abstract

Visual hallucination (VH) occurs when a multimodal large language model (MLLM) generates responses with incorrect visual details for prompts. Existing methods for generating VH test cases primarily rely on human annotations, typically in the form of triples: (image, question, answer). In this paper, we introduce *VHExpansion*, the first automated method for expanding VH test cases for MLLMs. Given an initial VH test case, VHExpansion automatically expands it by perturbing the question and answer through *negation* as well as modifying the image using both *common* and *adversarial perturbations*. Additionally, we propose a new evaluation metric, *symmetric accuracy*, which measures the proportion of correctly answered VH test-case pairs. Each pair consists of a test case and its negated counterpart. Our theoretical analysis shows that symmetric accuracy is an *unbiased evaluation metric* that remains unaffected by the imbalance of VH testing cases with varying answers when an MLLM is randomly guessing the answers, whereas traditional accuracy is prone to such imbalance. We apply VHExpansion to expand three VH datasets annotated manually and use these expanded datasets to benchmark seven MLLMs. Our evaluation shows that VHExpansion effectively identifies more VH test cases. Moreover, symmetric accuracy, being unbiased, leads to different conclusions about the vulnerability of MLLMs to VH compared to traditional accuracy metric. Finally, we show that fine-tuning MLLMs on the expanded VH dataset generated by VHExpansion mitigates VH more effectively than fine-tuning on the original, manually annotated dataset. We will publish code and data upon paper acceptance.

## 1 Introduction

Given a prompt containing both an image and a question, multimodal large language models (MLLMs) (Li et al., 2024b; Liu et al., 2023; Bai et al., 2023a; Li et al., 2023a; Tong et al., 2024a; Li et al., 2024a) generate a text response. MLLMs extend the capabilities of large language models (LLMs) (AI@Meta, 2024; Bai et al., 2023b; Touvron et al., 2023; Yang et al., 2024; Chiang et al., 2023) by enabling them to understand visual inputs. An MLLM typically comprises three main components: *a vision encoder*, *a vision-language connector*, and *an LLM*. Specifically, the vision encoder extracts visual embedding vectors from the image in the prompt, while the vision-language connector aligns these visual embedding vectors with the token-based input used by the LLM. The LLM then generates the text response based on the outputs of the vision-language connector and the text in the prompt. This integration allows MLLMs to tackle complex tasks like Visual Question Answering (VQA) (Tong et al., 2024b; Huang et al., 2024; Li et al., 2023b; Fu et al., 2023).

Despite significant advancements, MLLMs are prone to a critical flaw known as visual hallucination (VH) (Huang et al., 2024; Liu et al., 2024), where the model generates responses containing incorrect or misleading visual information. For example, Figure 1 illustrates a VH case where the MLLM provides an incorrect response regarding the number of spots on a butterfly's wings. VH can lead to catastrophic outcomes, especially in high-stakes applications such as autonomous driving (Wen et al., 2023; Chen et al., 2024), medical diagnostics (Qiu et al., 2022), and content moderation (Kumar et al., 2024). Therefore, VH poses significant obstacles to the safe deployments of MLLMs. This concern is highlighted in the U.S. Executive Order on Trustworthy AI (House, 2023), which emphasized rigorous testing of AI systems to identify and mitigate their potential harms. Therefore, developing methods to test and mitigate VH in MLLMs is crucial for ensuring their safety.

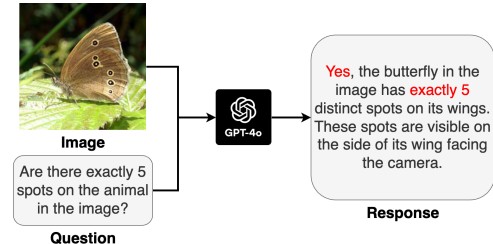

Figure 1: An example of visual hallucination (VH) in MLLM. The red text indicates the hallucinated response, since there are actually six spots on the butterfly's wings.

Existing VH testing relies on either manual (Li et al., 2023b) or semi-automated (Huang et al., 2024; Tong et al., 2024b) methods to construct test cases, both of which require extensive human annotations. As MLLMs evolve

rapidly, these methods struggle to scale up VH testing, limiting the number of test cases and thus hindering comprehensive testing of MLLMs' vulnerability to VH. Furthermore, existing VH testing methods do not consider adversarial testing (Goodfellow et al., 2014; Carlini & Wagner, 2017) in a white-box setting, where an adversary with full knowledge of the target MLLM can craft adversarial examples to trigger VH through adding human-imperceptible perturbations to the images. This is particularly relevant for open-sourced MLLMs whose model parameters are public. Thus, automated and adversarial methods for generating VH test cases are urgently needed.

**Our work:** To address these challenges, we introduce VHExpansion, the first *automated* framework to generate VH test cases for MLLMs. Given an initial VH test case, VHExpansion generates additional ones using a combination of *negation* as well as *common* and *adversarial image perturbations*. Each VH test case is a VQA triple consisting of an image, a question, and a ground-truth answer. Negation flips the question and answer. To automate negation, we leverage an LLM with a specifically designed prompt. For common image perturbations, we process the image via frequently encountered image processing operations such as JPEG compression, Gaussian noise, etc.. For adversarial image perturbations, we add a human-imperceptible perturbation to the image so that the resulting embedding vector, generated by the vision encoder, differs significantly from the original. We formulate finding the perturbation as a constrained optimization problem, solved using Projected Gradient Descent (Madry et al., 2018) or the iterative Fast Gradient Sign Method (Kurakin et al., 2018). We apply VHExpansion to expand three existing VH datasets annotated manually and use these expanded datasets to benchmark seven MLLMs. Our evaluation demonstrates that VHExpansion effectively identifies more VH test cases.

Moreover, we introduce a new evaluation metric called *symmetric accuracy*, which measures the proportion of correctly answered VH test-case pairs, where each pair includes a VH test case and its negated counterpart. Symmetric accuracy captures the consistency of an MLLM in accurately answering both the original and negated questions. In fact, we theoretically show that symmetric accuracy is an *unbiased evaluation metric* that remains unaffected by the imbalance of VH testing cases with varying answers when an MLLM is randomly guessing the answers, whereas traditional accuracy is prone to such imbalance. Our empirical benchmark results show that symmetric accuracy and traditional accuracy can lead to different conclusions about MLLMs' vulnerability to VH. For instance, on the POPE dataset (Li et al., 2023b), Cambrian-1 (Tong et al., 2024a) achieves a higher traditional accuracy than LLaVA-NeXT (Li et al., 2024a) (0.887 vs. 0.879), but performs worse in symmetric accuracy (0.745 vs. 0.798).

Finally, we demonstrate that fine-tuning an MLLM on the expanded VH test cases generated by VHExpansion significantly mitigates visual hallucinations. For example, our experiments show that when fine-tuning LLaVA-1.5 on the POPE dataset, using randomly sampled 200 VH test cases results in a symmetric accuracy of 0.180, whereas fine-tuning on both the sampled VH test cases and the corresponding expanded 1,800 more VH test cases achieves a symmetric accuracy of 0.711. This highlights the effectiveness of VHExpansion in mitigating VH in MLLMs. Additionally, our evaluation shows that fine-tuning does not compromise the model's performance on other general-purpose VQA datasets, preserving its broader functionality.

To summarize, we make the following contributions in this work:

- We introduce VHExpansion, the first automated framework for generating VH test cases in MLLMs, combining negation and common and adversarial image perturbations.

- We propose a new evaluation metric, symmetric accuracy, to quantify an MLLM's performance. Symmetric accuracy is unaffected by the imbalance of test cases when an MLLM makes random guessing.

- We demonstrate that fine-tuning MLLMs on the expanded test cases generated by VHExpansion significantly mitigates VH while maintaining general performance on other VQA datasets.

## 2 RELATED WORK

### 2.1 MLLMs

MLLMs (Li et al., 2024b; Liu et al., 2023; Bai et al., 2023a; Li et al., 2023a; Tong et al., 2024a; Li et al., 2024a) have revolutionized the ability of LLMs to respond to prompts containing images and questions. Recall that MLLMs typically comprise three components: a vision encoder, a vision-language connector, and an LLM. Vision encoders are often pre-trained via self-supervised learning (Radford et al., 2021; Oquab et al., 2023) on large datasets of unlabeled images or image-text pairs. Among the widely used vision encoders are those from the CLIP family (Radford et al., 2021), including CLIP-ViT-L/14 (Radford et al., 2021), EVA-CLIP ViT-g/14 (Fang et al., 2023), and OpenCLIP ConvNeXt-XXL (Ilharco et al., 2021; Liu et al., 2022). Cambrian-1 (Tong et al., 2024a) also incorporates other vision encoders, including DINOv2 ViT-L/14 (Oquab et al., 2023) and SigLIP ViT-SO400M/14 (Zhai et al., 2023). Recently, several types of vision-language connectors have been introduced, such as 2-layer multilayer perceptrons (MLPs), Q-Former (Dai et al., 2023), 1-layer cross-attention mechanisms (Bai et al., 2023b), and Spatial Visual Aggregator (Tong et al., 2024a). The backbone LLMs used in MLLMs can

Figure 2: Overview of VHExpansion. Green text and boxes indicate text and images modified by VHExpansion.

be models like Llama2 (Touvron et al., 2023), Llama3 (AI@Meta, 2024), Vicuna (Chiang et al., 2023), and Qwen (Bai et al., 2023a).

## 2.2 METHODS TO GENERATE VH TEST CASES

To detect and mitigate VH in MLLMs, several methods to generate VH test cases (Li et al., 2023b; Huang et al., 2024; Tong et al., 2024b; Guan et al., 2023) have been proposed. These methods can be categorized into two types: *manual* and *semi-automatic*. Manual methods (Li et al., 2023b; Guan et al., 2023) involve creating each VH test case through human effort. For example, POPE (Li et al., 2023b) requires human annotation for each image to identify the objects within it and then design corresponding questions based on these objects, including some randomly introduced non-existent objects. Note that the images in POPE are also human-created.

To reduce the human labor involved in generating VH test cases, semi-automatic methods (Huang et al., 2024) have been developed. For instance, VHTest uses GPT-4V (Achiam et al., 2023) and DALL·E-3 (Betker et al., 2023) to facilitate construction of VH test cases. Specifically, it first employs CLIP (Radford et al., 2021) and DINO (Oquab et al., 2023) to detect images from benchmark datasets that may trigger VH in MLLMs. These images are then passed to GPT-4V to generate textual descriptions. The generated text descriptions are subsequently passed to DALL·E-3 to create more images. Based on these AI-generated images, human workers manually identify objects within them and design questions, along with the corresponding ground-truth answers. Similarly, MMVP (Tong et al., 2024b) uses CLIP and DINO to identify image pairs that have a high CLIP score but a low DINO score. Human workers then manually examine the differences between these paired images and formulate questions/answers based on those differences.

## 2.3 MITIGATING VH VIA FINE-TUNING

With VH datasets constructed by these methods, MLLMs can be fine-tuned on them to mitigate VH (Huang et al., 2024). This approach enables the MLLMs to learn from instances of VH, allowing them to distinguish between accurate visual representations and hallucinated content. By exposing the models to diverse VH instances during fine-tuning, they can better generalize and reduce the occurrence of hallucinations (Huang et al., 2024).

## 3 OUR VHEXPANSION

Figure 2 shows an overview of our VHExpansion. Given an initial VH test case, VHExpansion automatically generates additional VH test cases by modifying the question and answer through negation, as well as modifying the image through common and adversarial image perturbations. We denote a VH test case as $\{x_I, x_Q, y_A\}$, where $x_I$ and $x_Q$ are respectively the image and text question in the prompt, while $y_A$ is the ground-truth answer. To support automated evaluation, we focus on binary questions in this work, i.e., $y_A$ is either "yes" or "no". Note that non-binary question-answer pairs $(x_Q, y_A)$ can be rewritten as binary counterparts.

### 3.1 MODIFYING QUESTION $x_Q$ AND ANSWER $y_A$ VIA NEGATION

Given a VH test case $\{x_I, x_Q, y_A\}$, the goal of negation is to transform it into $\{x_I, \neg x_Q, \neg y_A\}$. Our VHExpansion automates this process using an LLM with a custom prompt (showed in Figure 3). This prompt takes $x_Q$ as input and instructs the LLM to output a negated question using predefined transformation rules, such as adding negation prefixes or modifying key words to reverse the meaning of $x_Q$.

---

**Negation Prompt**

Rephrase the following question to be a negated question for the original question. The rephrase method is to add prefix 'Is it false' before the original question in a declarative sentence or change all occurrences of the "a/an" to "no" for simple cases. Below are the rules must be followed when rephrasing the question: DO NOT CHANGE OR ADD ANY INFORMATION to the sentence, such as the case of any letters except the first letter of the sentence, tenses, the order of clauses, pronouns, etc.. You should only return the rephrased question. The question is: $[x_Q]$.

---

Figure 3: Prompt used to instruct an LLM to negate a question $x_Q$.

The primary intuition behind negation is that an MLLM may simply guess the answer (i.e., "yes" or "no") correctly for binary questions without really understanding the image. In particular, some MLLMs such as LLaVA-1.5 tend to answer "yes" for binary questions (Liu et al., 2023). Therefore, if the VH test cases are imbalanced and a majority of them have "yes" as ground-truth answers, such MLLMs would have high *accuracy* without understanding the images, misleading developers to think that the MLLMs are not vulnerable to visual hallucination. However, such MLLMs would be likely to answer incorrectly for the negated questions, leading to low accuracy on them. Thus, the VH test cases and their negated versions can better quantify the vulnerability of an MLLM to visual hallucination. In fact, in Section 4, we propose a new evaluation metric, called symmetric accuracy, which measures the percentage of correctly answered VH test-case pairs, each of which includes a test case and its negated version. In Section 4, we theoretically show that symmetric accuracy is unaffected by the imbalance of VH test cases with answers "yes" and "no" when the MLLM makes random guessing, while accuracy on the original VH test cases alone is prone to such imbalance.

## 3.2 MODIFYING IMAGE $X_I$

**Common image perturbations:** In real-world scenarios, images often undergo standard editing operations for various purposes. For example, images are frequently compressed using formats like JPEG to reduce transmission costs over the Internet. These image edits are known as *common image perturbations* (Hendrycks & Dietterich, 2019). Our VHExpansion uses these perturbations to generate additional VH test cases. Given a VH test case $\{x_I, x_Q, y_A\}$, we apply a common perturbation method $\mathcal{T}$ to the image $x_I$, creating a new VH test case $\{\mathcal{T}(x_I), x_Q, y_A\}$. The intuition is that for a slightly perturbed image $\mathcal{T}(x_I)$, the ground-truth answer $y_A$ should remain unchanged for the same question $x_Q$. However, this subtle alteration may trigger VH in an MLLM. We focus on four common image perturbations: Gaussian Noise, Brightness Adjustments, Defocus Blur, and JPEG Compression. Further details on these common perturbations are provided in Section C of the Appendix.

**Adversarial image perturbations:** In the context of adversarial image perturbations, we consider a white-box setting where an adversary, with full knowledge of the target MLLM's model parameters, crafts nearly-imperceptible adversarial perturbations to generate VH test cases. Given an original VH test case $\{x_I, x_Q, y_A\}$, the adversarial image perturbation generates a new test case $\{x_I + \delta^*, x_Q, y_A\}$, where $\delta^*$ is the adversarial perturbation. Our intuition is that for a VH test case that does not trigger VH in an MLLM $M$, VHExpansion creates perturbations that cause the projected visual embedding vector from the vision-language connector to differ from the original. Conversely, if the test case already triggers VH in $M$, VHExpansion generates perturbations that make the projected visual embedding vector similar to the original. Formally, for an MLLM $M$ with vision encoder $M_E$ and vision-language connector $M_C$, we formulate finding $\delta^*$ as the solution to the following constrained optimization problem:

$$\delta^* = \begin{cases} \arg\min_\delta \left(- \cos\left(M_E \circ M_C(x_I), M_E \circ M_C(x_I + \delta)\right)\right), & \text{if } x_I \text{ does not trigger VH,} \\ \arg\min_\delta \left(\cos\left(M_E \circ M_C(x_I), M_E \circ M_C(x_I + \delta)\right)\right), & \text{if } x_I \text{ triggers VH,} \end{cases}$$
$$\text{s.t.} \quad ||\delta||_\infty \leq \epsilon, \tag{1}$$

where $M_E \circ M_C$ denotes the concatenation of the vision encoder and the vision-language connector, $\cos$ denotes cosine similarity, and $\epsilon$ is the $\ell_\infty$-norm constraint on the perturbation $\delta$ added to the image $x_I$. Note that when the VH test case already triggers VH, we initialize $\delta$ to be a non-zero vector with random value and apply early stopping to avoid the optimization result to be identical with the original image input $x_I$; and when the VH test case does not trigger VH, we initialize $\delta$ to be zero. Our algorithm solves the optimization problem in Equation 1 using either Projected Gradient Descent (PGD) (Madry et al., 2018) or the iterative Fast Gradient Sign Method (I-FGSM) (Kurakin et al., 2018). PGD iteratively updates $\delta$ via gradient ascent: $\delta = \delta - \gamma \cdot \nabla_\delta l$, where $l = \cos\left(M_E \circ M_C(x_I), M_E \circ M_C(x_I + \delta)\right)$, followed by projecting $\delta$ onto the feasible region using $\delta = \text{clip}(\delta, -\epsilon, \epsilon)$. I-FGSM differs from PGD by using the sign of the gradient instead: $\delta = \delta - \gamma \cdot \text{sign}(\nabla_\delta l)$.

Table 1: Statistics of existing VH datasets manually annotated.

| Dataset | # Images | # VH Test Cases |
|---------|----------|-----------------|
| MMVP (Tong et al., 2024b) | 300 | 300 |
| VHTest (Huang et al., 2024) | 650 | 1,200 |
| POPE (Li et al., 2023b) | 500 | 9,000 |

## 4 THEORETICAL ANALYSIS

In this section, we theoretically analyze the standard accuracy metric and our proposed symmetric accuracy metric for evaluating an MLLM model's performance when the model is making random guessing. Suppose we are given a VH test case $t = \{x_I, x_Q, y_A\}$, sampled from the distribution $\mathcal{T}$ of VH test cases, i.e., $t \sim \mathcal{T}$. Our analysis focuses on binary questions, i.e., $y_A$ is either "yes" or "no". Specifically, we denote by $q$ the probability that a randomly sampled $t$ has a ground-truth answer "yes". In other words, a randomly sampled $t$ has a ground-truth answer "no" with probability $1 - q$. $q$ quantifies the imbalance of the VH test cases with answers "yes" and "no".

We denote by $f$ an MLLM model and $f(x_I, x_Q)$ the MLLM's answer for the VH test case. $f(x_I, x_Q) \neq y_A$ indicates that the MLLM hallucinates. When the MLLM model makes random guessing to answer the test case without understanding the image $x_I$ and question $x_Q$, it outputs an answer "yes" or "no" randomly. Suppose the MLLM model guesses "yes" with probability $p$ and "no" with probability $1 - p$.

An evaluation metric measures the performance of an MLLM model $f$ on the VH test cases whose distribution is $\mathcal{T}$. Specifically, an evaluation metric takes $\mathcal{T}$ and $f$ as input and outputs a number (e.g., between 0 and 1), with a smaller number indicating that $f$ is more vulnerable to VH test cases from the distribution $\mathcal{T}$. An evaluation metric is *unbiased* if it does not depend on the imbalance of the VH test cases when the model $f$ makes random guessing, i.e., it does not depend on $q$. Otherwise, the evaluation metric is biased. Formally, we have the following definition.

**Definition 1** (Unbiased Evaluation Metric). *An evaluation metric is said to be unbiased if does not depend on $q$ when the MLLM model makes random guessing.*

Next, we formally define accuracy and prove that accuracy is a biased evaluation metric.

**Definition 2** (Accuracy). *Accuracy is the probability that an MLLM model $f$ correctly answers a VH test case $t = \{x_I, x_Q, y_A\}$ sampled from $\mathcal{T}$. Formally, we have: accuracy $= Pr_{t \sim \mathcal{T}}(f(x_I, x_Q) = y_A)$.*

**Theorem 1.** *Accuracy is a biased evaluation metric when $p \neq \frac{1}{2}$, where $p$ is the probability that the MLLM model guesses answer "yes".*

*Proof.* Please refer to Section A in Appendix. □

The above theorem shows that accuracy of an MLLM model depends on $q$ once it does not guess uniformly at random, and thus can be artificially inflated by random guessing, leading to misleading conclusions on an MLLM's vulnerability to visual hallucination. To address this limitation, we propose a new metric called symmetric accuracy. Formally, it is defined as follows:

**Definition 3** (Symmetric Accuracy). *Symmetric accuracy is the probability that an MLLM model $f$ correctly answers a VH test case $t = \{x_I, x_Q, y_A\}$ sampled from $\mathcal{T}$ and its negated version. Formally, we have: symmetric accuracy $= Pr_{t \sim \mathcal{T}}(f(x_I, x_Q) = y_A \wedge f(x_I, \neg x_Q) = \neg y_A)$.*

We prove that symmetric accuracy is an unbiased evaluation metric in the following theorem:

**Theorem 2.** *Symmetric accuracy is an unbiased evaluation metric.*

*Proof.* Please refer to Section B in Appendix. □

## 5 EXPERIMENTS

### 5.1 EXPERIMENTAL SETUP

**VH datasets:** We use three popular VH datasets: MMVP (Tong et al., 2024b), VHTest (Huang et al., 2024), and POPE (Li et al., 2023b). MMVP and VHTest consist of VH test cases across various object properties in images, such as color, counting, and position. In contrast, POPE focuses on VQA test cases related to existence

Table 2: Details of MLLMs.

| MLLM | Vision Encoder | Connector | LLM |
|---|---|---|---|
| LLaVA-1.5 (Liu et al., 2023) | CLIP-ViT-L/14 (Radford et al., 2021) | 2-layer MLP | Llama2-7B (Touvron et al., 2023) |
| InstructBLIP (Dai et al., 2023) | EVA-CLIP ViT-g/14 (Fang et al., 2023) | Q-Former (Li et al., 2023a) | Vicuna-7B (Chiang et al., 2023) |
| Qwen-VL-Chat (Bai et al., 2023b) | OpenCLIP ViT-bigG (Ilharco et al., 2021) | 1-layer Cross-Attention | Qwen-7B (Bai et al., 2023a) |
| LLaVA-NEXT (Li et al., 2024a) | CLIP-ViT-L/14 | 2-layer MLP | Llama3-8B (AI@Meta, 2024) |
| LLaVA-OneVision (Li et al., 2024b) | SigLIP ViT-SO400M/14 (Zhai et al., 2023) | 2-layer MLP | Qwen2-7B (Yang et al., 2024) |
| Cambrian-1 (Tong et al., 2024a) | CLIP ViT-L/14 SigLIP ViT-SO400M/14 OpenCLIP ConvNeXt-XXL (Liu et al., 2022) DINOv2 ViT-L/14 (Oquab et al., 2023) | Spatial Visual Aggregator (Tong et al., 2024a) | Llama3-8B |
| GPT-4o (OpenAI, 2024) | - | - | - |

Table 3: Accuracy, symmetric accuracy, and # new successful VH test cases for seven MLLMs on the three VH datasets.

(a) MMVP dataset

| MLLM | LLaVA-1.5 | InstructBLIP | Qwen-VL-Chat | Cambrian-1 | LLaVA-NeXT | LLaVA-OneVision | GPT-4o |
|---|---|---|---|---|---|---|---|
| Accuracy | 0.638 | 0.533 | 0.607 | 0.649 | 0.697 | 0.717 | 0.813 |
| Symmetric Accuracy | 0.356 | 0.320 | 0.210 | 0.268 | 0.430 | 0.333 | 0.663 |
| # New Successful VH test cases | 145 | 166 | 175 | 178 | 126 | 152 | 85 |

(b) VHTest dataset

| MLLM | LLaVA-1.5 | InstructBLIP | Qwen-VL-Chat | Cambrian-1 | LLaVA-NeXT | LLaVA-OneVision | GPT-4o |
|---|---|---|---|---|---|---|---|
| Accuracy | 0.542 | 0.499 | 0.537 | 0.631 | 0.588 | 0.632 | 0.709 |
| Symmetric Accuracy | 0.308 | 0.117 | 0.156 | 0.260 | 0.287 | 0.328 | 0.423 |
| # New Successful VH test cases | 599 | 643 | 627 | 670 | 585 | 647 | 523 |

(c) POPE dataset

| MLLM | LLaVA-1.5 | InstructBLIP | Qwen-VL-Chat | Cambrian-1 | LLaVA-NeXT | LLaVA-OneVision | GPT-4o |
|---|---|---|---|---|---|---|---|
| Accuracy | 0.861 | 0.860 | 0.692 | 0.887 | 0.879 | 0.889 | 0.861 |
| Symmetric Accuracy | 0.468 | 0.444 | 0.354 | 0.745 | 0.798 | 0.843 | 0.425 |
| # New Successful VH test cases | 3,978 | 4,368 | 4,845 | 2,046 | 1,281 | 976 | 4,504 |

VH, specifically identifying whether an object is present in an image. Table 1 summarizes the key statistics of these datasets. Note that for VHTest and POPE, a single image can be used in multiple VH test cases.

**MLLMs:** In our experiments, we evaluate seven MLLMs in total. In particular, six of these models are open-source, including LLaVA-1.5 (Liu et al., 2023), InstructBLIP (Dai et al., 2023), Qwen-VL-Chat (Bai et al., 2023b), LLaVA-NeXT (Li et al., 2024a), LLaVA-OneVision (Li et al., 2024b), and Cambrian-1 (Tong et al., 2024a), alongside one closed-source model, GPT-4o (OpenAI, 2024). These MLLMs demonstrate state-of-the-art performance across various VQA benchmarks and have diverse model architectures. Table 2 shows details of these MLLMs.

**Evaluation metrics:** We use accuracy and symmetric accuracy as our evaluation metrics, both of which are formally defined in Section 4. Our theoretical analysis demonstrates that symmetric accuracy is an unbiased evaluation metric, whereas traditional accuracy is biased. In our experiments, we illustrate how symmetric accuracy leads to different conclusions about the vulnerability of MLLMs to VH compared to the traditional accuracy metric. Subsequently, we use symmetric accuracy as our default evaluation metric unless otherwise mentioned. We also report the number of successful VH test cases generated by our VHExpansion.

**Parameter settings:** Unless otherwise mentioned, we use LLaVA-1.5 on MMVP dataset by default. We use GPT-4o as the LLM to negate all questions in VH test cases due to its state-of-the-art performance. We use the default parameter settings for all MLLMs. For common image perturbations, the parameters are set as follows: Gaussian Noise standard deviation $\sigma = 0.08$, Brightness Hue-Saturation -Value space constant $c = 0.5$, Defocus Blur radius $r = 5$, and JPEG Compression quality factor $q = 30$. More details of these common perturbations are shown in Section C in Appendix. For adversarial image perturbations, the default setting is: $\ell_\infty$-norm constraint $\epsilon = 8/255$, with 500 epochs for non-hallucinated VH test cases and 100 epochs for hallucinated test cases. In hallucinated test cases, each pixel of the initial perturbation is set to $5/255$ or $-5/255$ uniformly at random.

## 5.2 EXPERIMENTAL RESULTS

**Symmetric accuracy v.s. accuracy:** Table 3 shows accuracy and symmetric accuracy of the seven MLLMs across the three datasets MMVP, VHTest, and POPE. We have three main observations. First, symmetric accuracy reveals different conclusions about MLLM vulnerability to VH compared to traditional accuracy. For example, on the POPE dataset, Cambrian-1 has higher traditional accuracy than LLaVA-NeXT (0.887 vs. 0.879) but performs worse in symmetric accuracy (0.745 vs. 0.798). Second, when comparing symmetric accuracy across MLLMs, GPT-4o achieves the highest scores on MMVP and VHTest, particularly on MMVP with 0.663, indicating it is less prone to visual hallucinations than other models. LLaVA-OneVision scores the highest symmetric accuracy on POPE (0.843), likely due to its fine-tuning on simpler existence-based questions and possible overlap between POPE and its training data. Third, across VH datasets, all MLLMs perform worse on VHTest, with InstructBLIP scoring only 0.117. This is likely because VHTest contains VH test cases with AI-generated images and more complex questions that the models have not trained on, making it more challenging. In addition, expanding the dataset using negation allows us to generate more new VH instances, providing additional training data for fine-tuning, which leads to more robust models.

**Common and adversarial image perturbations generate more VH test cases:** Table 4 shows the symmetric accuracy on three datasets of different MLLMs before and after common image perturbations. We observe that symmetric accuracy slightly decreases after common image perturbations in most cases. This shows that most MLLMs are generally robust against common perturbations. For instance, LLaVA-NeXT's symmetric accuracy on the VHTest dataset drops from 0.287 to 0.272 under Gaussian Noise. However, there are still some notable exceptions. For example, Defocus Blur significantly reduces LLaVA-OneVision's accuracy on POPE, from 0.843 to 0.646; while three of four common perturbations even increase InstructBLIP's symmetric accuracy on VHTest.

Table 5 shows the symmetric accuracy on the three datasets of different MLLMs before and after adversarial image perturbations. Our main observation is that adversarial perturbations cause significant drops in symmetric accuracy for all MLLMs. For example, LLaVA-1.5's symmetric accuracy on POPE drops sharply from 0.468 to 0.017 when performing I-FGSM to craft adversarial perturbations. When comparing I-FGSM with PGD, I-FGSM consistently results in a larger decrease in accuracy, indicating it is more effective.

To conclude, MLLMs are fairly robust to common image perturbations but remain vulnerable to adversarial ones, highlighting the need for more adversarially robust training strategies. Furthermore, both common and adversarial perturbations lead to the generation of more new VH instances, with adversarial perturbations producing more VH instances than common perturbations, since MLLMs are more vulnerable to them.

**Manual verification for negation:** The correctness of our proposed symmetric accuracy metric relies on the validity of the negated questions, which are generated by LLMs. Since LLMs may exhibit hallucinations, these negated questions might not always be the negated counterparts of the original questions. Thus, it is necessary to verify whether the negated questions generated by the LLM are correct.

To validate the correctness of these negated questions, we randomly sampled 200 VQA triples (100 original-negation pairs) from each of the MMVP, VHTest, and POPE datasets, which were evaluated by four independent annotators. The task of the annotators was to judge whether each negated question was the correct negation of the corresponding original question generated by the LLM. The annotators unanimously agreed that all negated questions were correctly generated by the LLM. This result demonstrates the reliability of the LLM in generating valid negations.

## 5.3 ABLATION STUDY

We conduct a comprehensive ablation study on adversarial image perturbation using I-FGSM, since it is the most effective method to generate successful VH test cases in our VHExpansion.

**Impact of $\ell_\infty$-norm constraint $\epsilon$:** Recall that I-FGSM projects the perturbation into the feasible region defined by the $\ell_\infty$-norm constraint $\epsilon$ at each iteration. Table 6a shows the effect of varying $\epsilon$ on symmetric accuracy. We observe that symmetric accuracy initially decreases and then stabilizes as the $\ell_\infty$-norm constraint $\epsilon$ increases. For example, at $\epsilon = 4/255$, symmetric accuracy is 0.080, dropping to 0.051 at $\epsilon = 8/255$, after which it converges. This trend occurs because larger perturbations changes the visual embedding vector more significantly of an image for a non-hallucinated VH test case, which is more likely to trigger VH and thereby reducing symmetric accuracy.

**Impact of perturbation step size $\gamma$:** The perturbation step size $\gamma$ controls the update in every iteration of I-FGSM. Table 6b shows the impact of $\gamma$ on symmetric accuracy. We observe that symmetric accuracy is relatively insensitive to different small perturbation step size $\gamma$.

**Impact of iterations:** Since I-FGSM solves the optimization problem in Equation 1 iteratively, we study the impact of the number of iterations and present the results in Table 6c and Table 6d for hallucinated and non-hallucinated VH test cases, respectively. For hallucinated VH test cases, we observe that symmetric accuracy

Table 4: Symmetric accuracy and # new successful VH test cases on the three datasets of different MLLMs before and after common image perturbations. Due to API query limits, we sample 3,000 VH test cases from the POPE dataset for GPT-4o.

(a) MMVP dataset

| MLLM | LLaVA-1.5 | InstructBLIP | Qwen-VL-Chat | Cambrian-1 | LLaVA-NeXT | LLaVA-OneVision | GPT-4o |
|---|---|---|---|---|---|---|---|
| No Perturbation | 0.356 | 0.320 | 0.210 | 0.268 | 0.430 | 0.333 | 0.663 |
| Gaussian Noise | 0.353 | 0.187 | 0.147 | 0.213 | 0.370 | 0.317 | 0.643 |
| Brightness | 0.317 | 0.177 | 0.160 | 0.190 | 0.357 | 0.273 | 0.613 |
| Defocus Blur | 0.353 | 0.163 | 0.193 | 0.183 | 0.297 | 0.317 | 0.543 |
| JPEG Compression | 0.373 | 0.253 | 0.213 | 0.270 | 0.410 | 0.347 | 0.657 |

(b) # New successful VH test cases on MMVP dataset

| MLLM | LLaVA-1.5 | InstructBLIP | Qwen-VL-Chat | Cambrian-1 | LLaVA-NeXT | LLaVA-OneVision | GPT-4o |
|---|---|---|---|---|---|---|---|
| Gaussian Noise | 275 | 305 | 295 | 274 | 237 | 259 | 158 |
| Brightness | 270 | 300 | 309 | 270 | 238 | 273 | 165 |
| Defocus Blur | 256 | 305 | 296 | 283 | 255 | 262 | 185 |
| JPEG Compression | 253 | 299 | 285 | 279 | 233 | 239 | 148 |

(c) VHTest dataset

| MLLM | LLaVA-1.5 | InstructBLIP | Qwen-VL-Chat | Cambrian-1 | LLaVA-NeXT | LLaVA-OneVision | GPT-4o |
|---|---|---|---|---|---|---|---|
| No perturbation | 0.308 | 0.117 | 0.156 | 0.260 | 0.287 | 0.328 | 0.423 |
| Gaussian Noise | 0.312 | 0.138 | 0.170 | 0.258 | 0.272 | 0.279 | 0.429 |
| Brightness | 0.292 | 0.124 | 0.164 | 0.238 | 0.278 | 0.287 | 0.392 |
| Defocus Blur | 0.302 | 0.110 | 0.125 | 0.154 | 0.282 | 0.278 | 0.271 |
| JPEG Compression | 0.312 | 0.177 | 0.193 | 0.282 | 0.293 | 0.289 | 0.433 |

(d) # New successful VH test cases on VHTest dataset

| MLLM | LLaVA-1.5 | InstructBLIP | Qwen-VL-Chat | Cambrian-1 | LLaVA-NeXT | LLaVA-OneVision | GPT-4o |
|---|---|---|---|---|---|---|---|
| Gaussian Noise | 1,142 | 1,210 | 1,196 | 1,112 | 1,089 | 1,167 | 937 |
| Brightness | 1,160 | 1,209 | 1,212 | 1,114 | 1,079 | 1,139 | 1,001 |
| Defocus Blur | 1,162 | 1,210 | 1,210 | 1,133 | 1,086 | 1,145 | 1,164 |
| JPEG Compression | 1,168 | 1,216 | 1,220 | 1,075 | 1,069 | 1,129 | 938 |

(e) POPE dataset

| MLLM | LLaVA-1.5 | InstructBLIP | Qwen-VL-Chat | Cambrian-1 | LLaVA-NeXT | LLaVA-OneVision | GPT-4o |
|---|---|---|---|---|---|---|---|
| No perturbation | 0.468 | 0.444 | 0.354 | 0.745 | 0.798 | 0.843 | 0.425 |
| Gaussian Noise | 0.462 | 0.433 | 0.413 | 0.735 | 0.782 | 0.828 | 0.412 |
| Brightness | 0.444 | 0.428 | 0.345 | 0.738 | 0.757 | 0.819 | 0.389 |
| Defocus Blur | 0.449 | 0.435 | 0.486 | 0.699 | 0.789 | 0.646 | 0.396 |
| JPEG Compression | 0.465 | 0.440 | 0.402 | 0.726 | 0.828 | 0.724 | 0.410 |

(f) # New successful VH test cases on POPE dataset

| MLLM | LLaVA-1.5 | InstructBLIP | Qwen-VL-Chat | Cambrian-1 | LLaVA-NeXT | LLaVA-OneVision | GPT-4o |
|---|---|---|---|---|---|---|---|
| Gaussian Noise | 5,297 | 5,686 | 6,567 | 3,291 | 2,594 | 2,177 | 872 |
| Brightness | 5,502 | 5,788 | 7,662 | 3,221 | 2,634 | 2,337 | 916 |
| Defocus Blur | 5,429 | 5,727 | 5,410 | 4,025 | 3,049 | 3,605 | 897 |
| JPEG Compression | 5,260 | 5,685 | 6,909 | 3,173 | 2,535 | 2,878 | 896 |

remains consistently low as the number of iterations increases from 50 to 150. This is because I-FGSM updates the adversarial perturbations to increase the cosine similarity between the original and perturbed images for hallucinated VH test cases, maintaining the effectiveness of VH test cases. In non-hallucinated VH test cases, symmetric accuracy initially decreases and then stabilizes as the number of iterations increases from 100 to 900.

**Impact of repetition of evaluation:** Due to the inherent randomness in the decoding algorithm of MLLMs, we repeat the evaluation and report the average symmetric accuracy in Table 6e, varying the number of repetitions. We observe that symmetric accuracy remains consistent across different repetition counts, ranging from 0.040 to 0.051. This suggests that symmetric accuracy stabilizes after only a few repetitions, with even a single evaluation providing reliable results, thus avoiding unnecessary computational overhead.

**Impact of MLLM's temperature:** Temperature controls the randomness of MLLMs' responses, with higher temperatures typically leading to more diverse outputs. Table 6f shows the impact of temperature on LLaVA-1.5's symmetric accuracy. We observe a slight increase in symmetric accuracy as the temperature increases from 0 to 1, likely because the MLLM explores more diverse outputs at higher temperatures.

Table 5: Symmetric accuracy and # new successful VH test cases on the three datasets of different MLLMs before and after adversarial image perturbations. We cannot perform adversarial image perturbations for Cambrian-1 because of our limited GPU memory, and we do not have results for GPT-4o because it is closed-source.

(a) MMVP dataset

| Perturbation | LLaVA-1.5 | InstructBLIP | Qwen-VL-Chat | LLaVA-NeXT | LLaVA-OneVision |
|---|---|---|---|---|---|
| No perturbation | 0.356 | 0.320 | 0.210 | 0.430 | 0.333 |
| I-FGSM | 0.051 | 0.080 | 0.027 | 0.263 | 0.297 |
| PGD | 0.094 | 0.080 | 0.051 | 0.287 | 0.283 |

(b) # New successful VH test cases on the MMVP dataset

| Perturbation | LLaVA-1.5 | InstructBLIP | Qwen-VL-Chat | LLaVA-NeXT | LLaVA-OneVision |
|---|---|---|---|---|---|
| I-FGSM | 416 | 390 | 320 | 297 | 300 |
| PGD | 357 | 373 | 321 | 312 | 297 |

(c) VHTest dataset

| Perturbation | LLaVA-1.5 | InstructBLIP | Qwen-VL-Chat | LLaVA-NeXT | LLaVA-OneVision |
|---|---|---|---|---|---|
| No perturbation | 0.308 | 0.117 | 0.156 | 0.287 | 0.328 |
| I-FGSM | 0.102 | 0.053 | 0.097 | 0.144 | 0.147 |
| PGD | 0.166 | 0.059 | 0.117 | 0.204 | 0.249 |

(d) # New successful VH test cases on VHTest dataset

| Perturbation | LLaVA-1.5 | InstructBLIP | Qwen-VL-Chat | LLaVA-NeXT | LLaVA-OneVision |
|---|---|---|---|---|---|
| I-FGSM | 1,493 | 1,306 | 1,243 | 1,246 | 1,322 |
| PGD | 1,341 | 1,290 | 1,233 | 1,205 | 1,186 |

(e) POPE dataset

| Perturbation | LLaVA-1.5 | InstructBLIP | Qwen-VL-Chat | LLaVA-NeXT | LLaVA-OneVision |
|---|---|---|---|---|---|
| No perturbation | 0.468 | 0.444 | 0.354 | 0.798 | 0.843 |
| I-FGSM | 0.017 | 0.152 | 0.072 | 0.526 | 0.573 |
| PGD | 0.030 | 0.174 | 0.088 | 0.553 | 0.761 |

(f) # New successful VH test cases on POPE dataset

| Perturbation | LLaVA-1.5 | InstructBLIP | Qwen-VL-Chat | LLaVA-NeXT | LLaVA-OneVision |
|---|---|---|---|---|---|
| I-FGSM | 7,588 | 8,627 | 10,340 | 5,453 | 4,717 |
| PGD | 10,246 | 8,414 | 10,029 | 5,296 | 2,940 |

Table 6: Ablation study on symmetric accuracy for adversarial image perturbations for LLaVA-1.5 on MMVP dataset when using I-FGSM.

(a) $\ell_\infty$-norm constraint $\epsilon$

| $\epsilon$ | 4/255 | 8/255 | 12/255 | 16/255 |
|---|---|---|---|---|
| Symmetric Accuracy | 0.080 | 0.051 | 0.047 | 0.053 |

(b) Perturbation step size $\gamma$

| $\gamma$ | 0.3/255 | 0.4/255 | 0.5/255 | 0.6/255 | 0.7/255 |
|---|---|---|---|---|---|
| Symmetric Accuracy | 0.051 | 0.059 | 0.051 | 0.054 | 0.040 |

(c) Iterations for hallucinated VH test cases

| Iterations | 50 | 75 | 100 | 125 | 150 |
|---|---|---|---|---|---|
| Symmetric Accuracy | 0.049 | 0.062 | 0.051 | 0.042 | 0.054 |

(d) Iterations for non-hallucinated VH test cases

| Iterations | 100 | 300 | 500 | 700 | 900 |
|---|---|---|---|---|---|
| Symmetric Accuracy | 0.090 | 0.058 | 0.051 | 0.050 | 0.050 |

(e) Repetition of evaluation

| # Repetition | 1 | 2 | 3 | 4 | 5 |
|---|---|---|---|---|---|
| Symmetric Accuracy | 0.043 | 0.040 | 0.051 | 0.047 | 0.049 |

(f) Temperature of MLLM

| Temperature | 0.0 | 0.2 | 0.4 | 0.6 | 0.8 | 1.0 |
|---|---|---|---|---|---|---|
| Symmetric Accuracy | 0.037 | 0.051 | 0.066 | 0.099 | 0.104 | 0.096 |

## 5.4 MITIGATING VH VIA FINE-TUNING

Huang et al. (2024) demonstrate that fine-tuning MLLMs on VH datasets constructed using VH test case generation methods can help mitigate VH. In this section, we compare the symmetric accuracy across three scenarios: 1) before fine-tuning, 2) fine-tuning on original VH test cases generated by other methods, and 3) fine-tuning on original VH test cases generated by other methods combined with expanded VH test cases from our VHExpansion.

Table 7: Symmetric accuracy before and after fine-tuning on different image and VQA combinations.

| | Before Fine-tuning | After Fine-tuning on Original VH Test Cases | After Fine-tuning on Our Expanded VH Test Cases |
|---|---|---|---|
| **MMVP** | 0.207 | 0.172 | 0.343 |
| **VHTest** | 0.206 | 0.208 | 0.225 |
| **POPE** | 0.180 | 0.189 | 0.711 |

Table 8: Scores on MME Perception and MME Cognition before and after fine-tuning.

| | Before Fine-tuning | After Fine-tuning on Original VH Test Cases | After Fine-tuning on Our Expanded VH Test Cases |
|---|---|---|---|
| **MME Perception** | 1459.3 | 1456.7 | 1434.4 |
| **MME Cognition** | 335.4 | 327.5 | 323.9 |

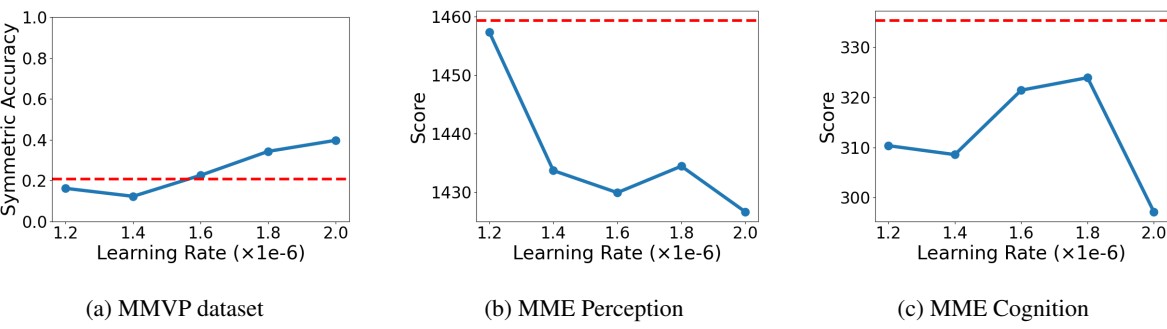

(a) MMVP dataset      (b) MME Perception      (c) MME Cognition

Figure 4: Impact of learning rate on symmetric accuracy for the MMVP dataset, and scores on MME perception and MME cognition, when fine-tuning LLaVA-1.5 on our expanded VH test cases. The red horizontal lines represent the performance of LLaVA-1.5 before fine-tuning.

**Experimental settings:** We use LLaVA-1.5 as the fine-tuning MLLM. For fine-tuning on the original VH test cases generated by other methods, we randomly sample 200 VH test cases from each of the MMVP, VHTest, and POPE datasets, along with 4,000 randomly sampled VQA triples from the LLaVA-1.5 fine-tuning data (Liu et al., 2023). For fine-tuning on our expanded VH test cases, we expand the previously sampled 200 VH test cases from each of the three datasets using negation and adversarial image perturbations, resulting in 800 VH test cases. To further increase data diversity, we use GPT-4o to rephrase the questions four times for each VH test case, generating four additional versions of each. Consequently, our expanded fine-tuning set contains 4,000 VH test cases and the sampled 4,000 VQA triples from the fine-tuning data of LLaVA-1.5. All remaining VH test cases from the three VH datasets, along with their adversarially perturbed versions, are used as evaluation data. Following LLaVA-1.5 (Liu et al., 2023), we fine-tune LLaVA-1.5 using LoRA (Hu et al., 2021) with a learning rate of $1.8 \times 10^{-6}$ for one epoch. All other parameters are set to the default fine-tuning settings of LLaVA-1.5.

**Experimental results:** The comparison results of fine-tuning are shown in Table 7 and Table 8. Our findings demonstrate that fine-tuning on our expanded VH test cases significantly improves symmetric accuracy across the three VH datasets. For instance, on the POPE dataset, symmetric accuracy increases slightly from 0.180 to 0.189 after fine-tuning on the original VH test cases, but rises substantially to 0.711 after fine-tuning on our expanded VH test cases. This highlights the effectiveness of using VH test cases generated by our VHExpansion to mitigate VH in MLLMs. Moreover, Table 8 shows that fine-tuning on our expanded VH test cases maintains the model's performance on other general-purpose VQA datasets, MME Perception and MME Recognition (Fu et al., 2023).

**Impact of fine-tuning learning rate:** Figure 4 illustrates the impact of different fine-tuning learning rates on symmetric accuracy for the MMVP dataset, scores on MME Perception and scores on MME Cognition. We observe that performance across these datasets is highly sensitive to the fine-tuning learning rate. At the learning rate of $1.8 \times 10^{-6}$, the fine-tuned MLLM achieves the best trade-off among performances on all three datasets.

## 6 CONCLUSION

In this paper, we introduce VHExpansion, an automated framework to generate VH test cases for MLLMs. VHExpansion significantly advances VH testing by automating the generation of test cases through techniques such as negation and image perturbations, both common and adversarial. We also propose an unbiased evaluation metric, symmetric accuracy, to measure the consistency of MLLMs in answering VH test cases and their negated counterparts. Our experiments demonstrate that, given VH test cases, VHExpansion can find more successful VH test cases. Importantly, fine-tuning MLLMs on the expanded VH test cases generated by VHExpansion significantly mitigates VH, while maintaining general performance on standard VQA tasks.

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

## A  PROOF OF THEOREM 1

*Proof.* Standard accuracy is defined as: Accuracy $= \Pr_{t \sim \mathcal{T}}(f(x_I, x_Q) = y_A)$. Since the model's predictions are independent of $y_A$ when random guessing:

$$E[\text{Accuracy}] = P(y_A = \text{Yes}) \cdot P(f(x_I, x_Q) = \text{Yes}) + P(y_A = \text{No}) \cdot P(f(x_I, x_Q) = \text{No})$$
$$= q \cdot p + (1 - q) \cdot (1 - p)$$
$$= 1 + (2p - 1) \cdot q - p.$$

This expression shows that $E[\text{Accuracy}]$ **depends** on the class distribution ($P(y_A = \text{Yes})$) if $p \neq \frac{1}{2}$. If the model's bias aligns with the majority class (e.g., $p$ is large when $P(y_A = \text{Yes})$ is large), $E[\text{Accuracy}]$ is artificially inflated, even though the model is merely guessing.

Therefore, standard accuracy is biased due to class imbalance and model bias. □

## B  PROOF OF THEOREM 2

*Proof.* Symmetric accuracy is defined as: Symmetric Accuracy $= \Pr_{t \sim \mathcal{T}}(f(x_I, x_Q) = y_A \wedge f(x_I, \neg x_Q) = \neg y_A)$. Since model predictions are independent of $y_A$ and independent between $x_Q$ and $\neg x_Q$ under random guessing:

$$E[\text{Symmetric Accuracy}] = P(f(x_I, x_Q) = y_A) \cdot P(f(x_I, \neg x_Q) = \neg y_A)$$
$$= P(y_A = \text{Yes}) \cdot P(f(x_I, x_Q) = \text{Yes}) \cdot P(f(x_I, \neg x_Q) = \text{No})$$
$$+ P(y_A = \text{No}) \cdot P(f(x_I, x_Q) = \text{No}) \cdot P(f(x_I, \neg x_Q) = \text{Yes})$$
$$= q \cdot p(1 - p) + (1 - q) \cdot p(1 - p)$$
$$= p(1 - p). \tag{2}$$

Therefore, $E[\text{Symmetric Accuracy}] = p(1 - p)$, which is **independent** of the class distribution ($P(y_A = \text{Yes})$). Thus, symmetric accuracy is an unbiased evaluation metric with respect to class imbalance.

□

## C  DETAILS OF COMMON IMAGE PERTURBATIONS

- **Gaussian Noise** In this method, Gaussian noise is randomly sampled from a distribution with zero mean and a standard deviation of $\sigma$. The image pixel values are first converted to the range [0, 1], and the generated noise is then added to these values. This process simulates the noise real-world images might experience during transmission. In our experiments, the standard deviation $\sigma$ is set to 0.08.

- **Brightness** This method adjusts image brightness by modifying its V (value) channel in the HSV color space. The input image is first normalized to [0, 1] and converted from RGB to HSV. The brightness is then altered by adding a constant $c$ to the V channel, with values clipped to the range [0, 1]. The image is finally converted back to RGB. In our experiments, the constant $c$ is set to 0.5.

- **Defocus Blur** This method applies a defocus blur to the image using a disk-shaped kernel. The input image is normalized to [0, 1], and a disk kernel of radius $c$ is generated. Each of the three RGB channels is filtered independently with this kernel, then recombined and clipped to the range [0, 1]. The radius $c$ is set to 5 in our experiments.

- **JPEG Compression** This method compresses the input image using a specified quality factor $q$. Lower $q$ values result in higher compression and more artifacts, while higher values retain more image quality. In our experiments, the quality factor $q$ is set to 30.

