# OpenReview forum: "Automatically Generating Visual Hallucination Test Cases for Multimodal Large Language Models"
_ICLR.cc/2025/Conference — ICLR 2025 Conference Withdrawn Submission_

### Official Review · Reviewer_25Gz · 2024-11-01

**Soundness:** 2
**Presentation:** 2
**Contribution:** 2
**Rating:** 5
**Confidence:** 3

**Summary:**

This paper introduces VHExpansion, an automated method for generating visual hallucination test cases for MLLMs. Each test case is generated by perturbing the question and answer through negation and modifying the image using common and adversarial perturbations. The paper also proposes an evaluation metric, named symmetric accuracy and shows it's unbiased.

**Strengths:**

1, The paper has done extensive ablation study and has evaluated different hyperparameters.
2, The paper is clearly written.

**Weaknesses:**

1, The paper mainly addresses the MLLM hallucination by constructing more data and then fine-tuning on them, which is not very exciting.
2, In Table.8, we can see the proposed method causes the MLLM comprehensive ability to decrease.

**Questions:**

1, In this paper, the authors stated that the proposed metric (symmetric accuracy) is better than the standard accuracy. Other than the theoretical analysis, are there any human evaluation/human preference that can further show that this metric is better preferred?
2, In table.7, can the authors provide more explanation about why on some dataset (POPE), the improvement is huge while on others it is not as significant.
3, Other than Llava-1.5, have the authors tried fine-tuning on different open source models?(Table.7) Are there also improvements in the performances?

---

### Official Review · Reviewer_D8XG · 2024-11-02

**Soundness:** 2
**Presentation:** 2
**Contribution:** 2
**Rating:** 3
**Confidence:** 4

**Summary:**

This paper introduces VHExpansion, an automated framework for generating visual hallucination (VH) test cases in multimodal large language models (MLLMs) using negation and image perturbations. It proposes a new metric, symmetric accuracy, to address bias in traditional accuracy, providing a more reliable measure of VH vulnerability. Experiments demonstrate that symmetric accuracy reveals more about MLLM susceptibility to VH, and fine-tuning with VHExpansion cases reduces hallucinations effectively.

**Strengths:**

(1) The authors identified a potential bias in the original yes/no response QA questions and proposed a symmetric accuracy evaluation metric, which is very insightful for current evaluations.

(2) The authors conducted comprehensive experiments, including a comparison between traditional metrics and the new metric, as well as improvements in model performance after fine-tuning.

(3) Traditional hallucination benchmarks mostly rely on manual construction, while automated generation methods greatly reduce the need for human labor.

**Weaknesses:**

(1) As an automated method for generating VH test cases, this approach addresses the issue of insufficient data volume in previous benchmarks like MMHAL-BENCH and POPE.

(2) The entire paper seems like an application of transferring an attack method to VLMs, lacking novelty, as the attack approach itself has been previously used.

(3) The overall paradigm of automatically constructing benchmarks is also very similar to many works, such as:
     1. Autobencher: Creating salient, novel, difficult datasets for language models
     2. Multimodal Self-Instruct: Synthetic Abstract Image and Visual Reasoning Instruction Using Language Model
     3. How Many Unicorns Are in This Image? A Safety Evaluation Benchmark for Vision LLMs
For example, in the third paper, some test cases were also constructed using a similar attack method.

**Questions:**

(1) Since the negation part modifies the question using LLM, has the quality of the modified questions been validated?

---

### Official Review · Reviewer_JWsb · 2024-11-03

**Soundness:** 3
**Presentation:** 3
**Contribution:** 2
**Rating:** 5
**Confidence:** 5

**Summary:**

This paper introduces VHExpansion, an automated method for expanding Visual Hallucination (VH) test cases in multimodal large language models (MLLMs). VH occurs when an MLLM generates responses with incorrect visual details. The current methods for generating VH test cases are primarily dependent on human annotations. VHExpansion, however, expands these test cases by altering the question and answer through negation and modifying the image using both common and adversarial perturbations.

The paper also proposes a new evaluation metric, symmetric accuracy, which measures the proportion of correctly answered VH test-case pairs, including a test case and its negated counterpart. The authors argue that symmetric accuracy is an unbiased metric, unaffected by the imbalance of VH testing cases with varying answers when an MLLM is randomly guessing the answers, unlike traditional accuracy.

The researchers applied VHExpansion to three VH datasets and used these expanded datasets to benchmark seven MLLMs. The results showed that VHExpansion effectively identifies more VH test cases and that symmetric accuracy provides a more unbiased evaluation of MLLMs' vulnerability to VH. The paper also demonstrates that fine-tuning MLLMs on the expanded VH dataset generated by VHExpansion is more effective in mitigating VH than fine-tuning on the original, manually annotated dataset.

**Strengths:**

+ The proposed test case generation method can automatically expand the number of VH data samples, reducing reliance on human annotations and enhancing scalability in testing multimodal large language models.
+ This paper considered the evaluation bias and propose an unbiased evaluation metric, offering a more reliable assessment compared to traditional accuracy metrics.
+ Experiments are extensive and sufficient.

**Weaknesses:**

- While the paper discusses common and adversarial perturbations, it might not fully explore the spectrum of possible perturbations that could trigger visual hallucinations in MLLMs, such as resize/affine transform/crop/generation-based image editting, etc. There could be more sophisticated or subtle perturbations not covered.
- The method relies on an LLM to automate negation of questions, which might introduce its own errors or biases. The makes the method still need correction by human and thus the method is still semi-automatic instead of the stated fully-automatic.
- In my opinion, Table 8 shows that fine-tuning on expanded VH test cases degrades the model’s
performance on other general-purpose VQA datasets (MME in this case) since the scores are harmed. However, authors tend to interpret the results as " the model’s performance are maintained" in line 565.

**Questions:**

See weakness.

---

### Official Review · Reviewer_VWMb · 2024-11-04

**Soundness:** 2
**Presentation:** 3
**Contribution:** 2
**Rating:** 5
**Confidence:** 4

**Summary:**

This paper introduces VHExpansion, an automated framework for generating additional visual hallucination (VH) test cases based on existing ones to evaluate multimodal large language models (MLLMs). Targeting binary questions, VHExpansion enhances current test cases by questions and answers through negation, and modifying images using both common and adversarial perturbations. The authors propose a new evaluation metric, symmetric accuracy, which serves as an unbiased measure of MLLM performance in addressing VH by assessing the model’s ability to correctly answer both original and negated questions. Experimental results show that VHExpansion effectively identifies more VH test cases and that fine-tuning MLLMs on the expanded dataset significantly reduces VH without compromising overall performance on other visual question answering (VQA) tasks.

**Strengths:**

The paper is well-written, with clearly defined concepts that are easy to understand. The methodology and experimental design are presented in detail, ensuring transparency and allowing readers to follow the research process with ease.

The authors enhance the rigor of their approach by conducting manual verification to confirm the correctness of the proposed negation operation.

Thorough experiments and comprehensive analyses effectively demonstrate the utility and impact of the VHExpansion framework. Moreover, the paper highlights how the newly introduced symmetric accuracy metric offers a more robust and unbiased evaluation of MLLM performance on VH test cases. This provides insights for future researchers aiming to tackle these challenges.

**Weaknesses:**

1. The proposed method's focus on binary questions limits its generalizability, as both the VHExpansion framework and the symmetric accuracy metric may not directly apply to other types of VH datasets with non-binary answer formats. It would be valuable to explore whether the performance improvements from fine-tuning on expanded VH test cases can transfer to other VH datasets with different answer structures, such as multiple-choice questions. Although the symmetric accuracy metric is tailored to binary questions, the image perturbation techniques could potentially be applied to other VH datasets.
2. As indicated in line 288, the VH datasets used in this study primarily focus on low-level understanding, such as object properties (e.g., color, counting, and position) and object presence. It would be interesting to evaluate whether the proposed method could be effective for VH scenarios related to more advanced comprehension, like sentiment or context-based analysis.
3. The paper would benefit from a more detailed analysis of question types. For example, while the authors note that *three of four common perturbations even increase InstructBLIP’s symmetric accuracy on VHTest,* they do not delve into the underlying reasons. Analyzing how the proposed method performs across different question types and the specific capabilities it tests would provide more insight. For instance, a question involving counting (as shown in Fig. 1) might become more challenging with defocus blur, whereas the question in Fig. 2 might benefit from the same blur by reducing visual distractions.
4. The experimental setup in the "MITIGATING VH VIA FINE-TUNING" section (5.4), claimed as a major contribution, has some issues. The comparison between "fine-tuning on the original VH test cases (baseline)" and "fine-tuning on expanded VH test cases using VHExpansion (expanded)" is not entirely fair. In the *baseline* condition, the fine-tuning dataset includes 200 VH samples and 4,000 general VQA data points, while the *expanded* condition features 4,000 VH samples alongside 4,000 general VQA data points. This imbalance skews the comparison, as the *expanded* condition naturally emphasizes VH cases more. A more balanced comparison would involve scaling (repeating) the VH questions in *baseline* up to match the 4,000 samples in the *expanded* condition, thus equalizing the training set size and ensuring a fair evaluation.
5. Additionally, the use of symmetric accuracy in the evaluation poses a challenge. The *baseline* condition does not include negated questions, yet the evaluation includes them. A fairer approach would be to assess performance using standard accuracy or conduct an ablation study that excludes negation questions in the *expanded* condition. This would clarify whether the observed improvements in VH performance are indeed due to the proposed method.



**Minor Points:**

- In Tables 3-5, the metric *# New successful VH test cases* is used. It would be beneficial to also provide the ratio of successful cases to the total number of generated questions for easier comparisons.

**Questions:**

1. The adversarial attack method employed appears somewhat outdated. Would incorporating more advanced techniques enhance the effectiveness of the proposed framework?
2. Could perturbations, such as defocus blur, degrade the image to a point where the question becomes unanswerable based on the available visual information?

---

### Note · Authors · 2024-11-14

**Comment:**

We sincerely thank all reviewers for their constructive feedback. We plan to incorporate these insights to improve the work for future submissions.

**Withdrawal Confirmation:**

I have read and agree with the venue's withdrawal policy on behalf of myself and my co-authors.